# Architectural Allostatic Overloading: Exploring a Connection between Architectural Form and Allostatic Overloading

**DOI:** 10.3390/ijerph20095637

**Published:** 2023-04-25

**Authors:** Cleo Valentine

**Affiliations:** Department of Architecture, University of Cambridge, Cambridge CB2 1PX, UK; crv29@cam.ac.uk

**Keywords:** architectural health, allostatic overloading, neuroimmunology, architectural neuroimmunology, urban health

## Abstract

This paper examines, conceptually, the relationship between stress-inducing architectural features and allostatic overload by drawing on literature from neuroimmunology and neuroarchitecture. The studies reviewed from the field of neuroimmunology indicate that chronic or repeated exposure to stress-inducing events may overwhelm the body’s regulatory system, resulting in a process termed allostatic overload. While there is evidence from the field of neuroarchitecture that short-term exposure to particular architectural features produce acute stress responses, there is yet to be a study on the relationship between stress-inducing architectural features and allostatic load. This paper considers how to design such a study by reviewing the two primary methods used to measure allostatic overload: biomarkers and clinimetrics. Of particular interest is the observation that the clinical biomarkers used to measure stress in neuroarchitectural studies differ substantially from those used to measure allostatic load. Therefore, the paper concludes that while the observed stress responses to particular architectural forms may indicate allostatic activity, further research is needed to determine whether these stress responses are leading to allostatic overload. Consequently, a discrete longitudinal public health study is advised, one which engages the clinical biomarkers indicative of allostatic activity and incorporates contextual data using a clinimetric approach.

## 1. Introduction

Over the last ten years, a number of studies have considered the impact of architecture on human health and wellness. Most of this research has examined the effects of direct, overt, and easily quantifiable factors, such as light exposure [1,2], noise levels [3,4], air pollution [5,6], water quality [7,8], and ambient temperature [9,10]. Facilitated by technological advances in virtual reality and biometric sensors, more recent studies have begun to empirically examine neurophysiological responses to the built environment. These findings indicate that visual exposure to certain subtle variations in the shape or configuration of the built environment, referred to in this paper as ‘architectural forms’, such as room width and wall curvature, can elicit neuroimmunological stress responses [11,12,13,14,15]. Neuroimmunological stress responses (‘stress responses’) are defined here as physiological “stress pathways that influence neurobiological (i.e., cellular activity) and neuroimmune (i.e., immune activity) function” [16] (p. 2).

The relationship between human health and chronic exposure to these architectural forms is yet to be fully understood. However, studies on social and environmental stress responses indicate that when a stress-inducing event is continuous or repetitive, the body’s regulatory system can become overwhelmed, and the physiological responses, which are initially protective, become damaging over time; this state is termed allostatic overloading [17,18,19,20,21,22,23]. Allostatic overloading, in turn, can lead to prolonged neural or neuroendocrine responses, which compromises the immune system, resulting in long-term damage to organs and tissues [24]. One reason for this is that dysregulated allostatic responses result in the sustained production of inflammatory antigens, which causes chronic systemic inflammation [25]. Chronic systemic inflammation has been identified as a contributing factor in the development of several diseases associated with high rates of disability and mortality, such as cardiovascular disease [26], cancer [27], chronic kidney disease [28], non-alcoholic fatty liver disease [29], and autoimmune [30] and neurodegenerative disorders [31]. Given what is known about both the impact of allostatic overloading on human health and the stress-eliciting responses of certain architectural forms, the impact of chronic exposure to these architectural forms on human health may be greater than previously understood.

This paper constitutes an interdisciplinary desk review of literature from neuroimmunology and neuroarchitecture. The paper has two primary aims (i) to understand, conceptually, the relationship between stress-inducing architectural features and allostatic loads (Section 3), and (ii) to outline how neuroarchitectural researchers might empirically measure architecturally mediated allostatic overload in the future (Section 4). The paper proceeds to delineate two principal approaches to assessing allostatic overload, namely, biometric analysis and clinimetrics. However, it is important to acknowledge that each of these methods has notable limitations. Specifically, there are concerns regarding the lack of consensus on measurement techniques, the limited scope of biomarkers, and the potential influence of confounding variables that hinder the isolation of spatial variables.

An important finding from this review is that while there is evidence that certain architectural features cause acute stress responses, the clinical biomarkers used in those studies differ to those used to measure allostatic load. Therefore, while the observed stress responses may suggest allostatic activity, it is not conclusive that those stress responses are resulting in allostatic overload without further empirical inquiry utilising the methods for studying allostatic load examined in Section 4. Section 5 provides insight into how these methods may be adopted in the field of neuroarchitecture.

## 2. Architecturally Mediated Allostatic Overloading

Recent studies have evidenced that exposure to certain architectural forms, such as room proportions [14,15], wall curvature [12,15], and window arrangement and size [11,13,14] regularly provoke stress responses in humans without their conscious perception. Conversely, an emerging body of empirical research has found that biophilic design has a restorative effect. Biophilic design refers to spatial designs which integrate live organic material [32]. Exposure to biophilic environments is found to enhance human health and well-being by eliciting ’stress-reducing’ responses within the body [33,34,35,36,37,38,39]. These beneficial responses are evidenced by higher frontal lobe alpha waves [34,36,40] and lesser frontal lobe activation [35,41], which indicates a more relaxed state of mind.

Similarly, research suggests that biomorphic design, or designs which “imitate the contours and motifs of organisms” or organic matter [32] (p. 16), may also produce physiological benefits [42,43,44]. Although there has been a limited number of studies on biomorphic spatial design [45], exposure to the associated fractal patterns has resulted in reductions in stress responses [46,47,48]. Salingaros and Masden (2008), for example, found that biologically inspired forms activate opioid receptors, which reduces pain and suppresses the release of stress-related endocrine hormones [49]. These findings suggest that in the presence of both biophilic and biomorphic architectural forms, we see evidence of stress downregulation. In short, architectural forms have the potential to either upregulate or downregulate stress responses. This paper is primarily concerned with upregulation (i.e., an increase in allostatic activity) and the potential for stress-inducing architectural forms to produce a cascade of allostatic events and result in allostatic overload.

Allostasis refers to the neural and neuroendocrine (i.e., hormonal) processes by which the body regulates stress [50]. For example, when presented with a physical or psychological threat, the amygdala sends a signal to the hypothalamus, which activates the adrenal glands to release epinephrine and cortisol into the bloodstream [51]. This results in a number of physiological changes: an elevated heart rate to pump more blood to the muscles, heart, and vital organs; increased blood pressure and breathing to draw more oxygen into the lungs; an increased supply of oxygen in the brain to heighten mental alertness and the release of blood sugar and fat stores for additional energy demands [51] (outlined in Figure 1 below). Consequently, the primary physiological mediators of allostasis (considered further in Section 3) include hormones of the hypothalamic-pituitary-adrenal (HPA) axis (the body’s primary stress response network), catecholamines (neurotransmitters responsible for the production of epinephrine, norepinephrine, and dopamine) and cytokines (immune cells partially responsible for regulating inflammatory responses) [51].

Put simply, allostatic processes are physiological responses to stress-inducing events which are designed to return the body to a state of equilibrium (i.e., homeostasis) by providing the necessary physiological responses to allow the individual to navigate the stressful situation. Therefore, allostatic processes have an important function in regulating the body over time and mitigating the damaging impacts of stressors. A stressor, in this context, is defined as a “threat, real or implied, to the psychological or physiological integrity of an individual” [50] (p. 108).

The research on allostasis establishes that when the body can quickly downregulate, as it is designed to do, these allostatic processes serve a protective function and are beneficial. However, if the stressor is chronic, then allostatic processes can overwhelm the body’s regulatory system and inhibit the allostatic process from returning the body to its stable state, thus resulting in allostatic overload. Only when there is prolonged exposure to a stressor does the cumulative effect begin to compromise the immune system, contributing to chronic systemic inflammation which, in turn, can increase one’s risk of developing certain diseases. However, it is unclear from the research presented to date precisely what the effect of prolonged or repeated exposure to stress-inducing architectural forms has on human health or whether such exposure results in allostatic overload.

In the short term, the allostatic response triggered by exposure to stress-inducing architectural forms appears to be either benign or within tolerable limits [11,12,13,14,15]. Whether the allostatic process results in negative health outcomes may be largely determined by the volume and duration of exposure. Therefore, architectural forms are not in and of themselves antagonistic to human health. The issue, of course, is that the stress-inducing architectural forms discussed above are rarely unique and seldom appear as a singular, isolated event. Instead, these forms are used liberally within the built environment. Furthermore, we tend to occupy spaces within the built environment either continuously or repeatedly, e.g., offices, homes, and schools. Therefore, architectural forms, such as windowless rooms or confined spaces, which are seemingly well-tolerated, may not be tolerable over extended periods of time.

Of particular concern is that exposure to the built environment is increasing at an exponential rate. By 2050, it is projected that two-thirds of the global population will live in cities [52]. On average, residents in developed nations spend over 90% of their time inside. These findings are illustrated in a 2007 study by Schweizer et al., who found that residents in Oxford, UK spent an average of 95.6% of their time indoors, with 66% of this time spent in their own homes. Moreover, vulnerable individuals in Europe (i.e., elderly residents, infants and young children, and people with compromised immunity) may spend up to 100% of their time indoors [53]. These numbers are expected to rise with the emergence of epidemiological conditions such as COVID-19 [54] and the increasing climatic volatility related to global warming [53].

In sum, viewing stress-inducing architectural forms through an allostatic-overloading framework suggests that chronic exposure to stress-inducing architectural forms may contribute to allostatic overloading, and subsequently, to the development of systemic inflammation. This process of architecturally mediated allostatic overloading may be exacerbated by the repetitive nature of architectural exposure, increasing urbanisation rates, and increased time spent indoors. While it is notably difficult to measure allostatic load, considerable progress has been made in measurement techniques. A body of research has emerged that examines the impact of aspects of the urban environment on allostatic load. Research on environmental stressors has established that factors such as lead exposure [55,56], perception of pollution [57], pollution [58], dangerous traffic [59], household crowding [60], environmental riskscapes [61], and poor indoor environmental quality [62] are all associated with a higher allostatic load.

Additionally, spatial epidemiologists have measured the relationship between allostatic load and neighbourhood characteristics [63]. These studies find that perceived neighbourhood quality, which often encompasses features of the built environment such as overall condition of residential units, availability of recreational space, and perceived disorder and deterioration of property, has the potential to affect allostatic load [64,65,66]. Greater tree cover within 500 m of residence was also linked with lower allostatic load [67]. While these studies have not considered the impact of architectural form on allostatic load, they provide insight into how we may begin to empirically study the impact of stress-inducing architectural forms, such as room proportions [14,15], wall curvature [12,15], and window arrangement and size [11,13,14] on allostatic load.

## 3. Measuring Allostatic Overloading

The physiological conditions associated with allostatic overloading have a complex aetiology. Studies measuring allostatic overload typically use a combined index comprised of indicators of cumulative stress on multiple organs and tissues. This section examines two primary methods of measuring allostatic load biomarkers and clinimetrics. In brief, biomarkers (Section 3.1) are objective measures of physiological responses, whereas clinimetrics (Section 3.2) combine these objective measures with subjective data, such as patient-reported symptoms. Each approach has their strengths and weaknesses, which are outlined below (Section 3.3).

### 3.1. Biomarker Assessment

The first method used to measure allostatic load engages a set of biological markers that are associated with chronic stress and measures those markers at multiple points in time to track changes in allostatic load. Biomarkers indicative of allostatic load have traditionally been divided into primary mediators and secondary mediators [68]. However, additional variables have since been incorporated (discussed below).

#### 3.1.1. Biomarkers

Primary mediators are defined as indicators of biochemical alterations within the neuroendocrine system that occur during the initiation of the stress response [63]. These mediators are either responsible for activating the stress response via the hypothalamic-pituitary-adrenal axis and the sympathetic-adrenal medullary axis, or they are a consequence of these primary cascades [63]. In their 2022 literature review, Beese et al. analysed 18 systematic reviews (a total of 238 studies reviewed) that evaluated the measurement of allostatic load in neighbourhoods. Their findings indicated that cortisol was the primary mediator most frequently employed, followed by epinephrine, norepinephrine, and dehydroepiandrosterone. In allostatic research, these variables are measured through blood serum (dehydroepiandrosterone) and urine (cortisol, epinephrine, norepinephrine) [69].

The secondary mediators pertain to the remodelling of receptor sites in the cardiovascular, immune, and metabolic systems that results from prolonged activation of the stress response [63]. These mediators serve to quantify the symptomatic expressions of a protracted stress response [63]. The secondary mediators most utilised were cardiovascular biomarkers, specifically those that measured systolic and diastolic blood pressure [63]. Regarding biomarkers associated with the metabolic system, high-density lipoprotein, glycosylated haemoglobin, total cholesterol, body mass index, and waist-to-hip ratio were the most frequently used [63].

Additional markers most frequently used in the research were the immune system biomarkers C-reactive protein, interleukin-6, and fibrinogen [63]. Several other cardiovascular biomarkers, including heart rate/pulse rate, pulse pressure, peak flow expiratory, and apolipoprotein A1, B, were infrequently employed and found in the literature [63]. Additionally, heart-rate variability and dopamine were found to have been used in relatively few studies over the years [63]. A comprehensive list of additional biomarkers is exhibited in Figure 2 below.

#### 3.1.2. Combined and Algorithmic Biomarker Indices

While single biomarkers are occasionally utilised to indicate alterations in allostatic activity within certain physiological systems (i.e., variations in cortisol may indicate deviations in neuroendocrine activity), composite biomarker measures that encompass combined assessments of multiple biomarkers are frequently employed [70]. One common technique to compute allostatic load involves uniformly assigning weightage to all biomarkers within a composite measure that signifies the summation of the measured biomarkers, with each biomarker categorised as “0” for normal/low values or “1” for high values, as per nationally standardised ranges [63]. High allostatic loads are ascertained by summing the score, and an elevated risk of high allostatic load is indicated when three or more biomarkers register high-risk readings [71,72].

The most commonly employed approach for calculating allostatic load utilises the extreme quantiles, such as the 10th and 90th percentile, of each biomarker to establish the acceptable range [63]. However, given the absence of a standardised quantile threshold, researchers typically specify the percentile values for individual biomarkers and justify the chosen cut-off point [72,73,74]. The utilisation of z-scores represents a third approach for calculating allostatic load, although this method is comparatively less prevalent in comparison to the two principal methods [63]. Lastly, Mahalanobis Distance Method [75,76] estimates allostatic load by utilising continuous biomarker data to calculate the statistical distance from the centre of the joint distribution of biomarkers, which accounts for the covariance among them.

**Figure 2 ijerph-20-05637-f002:**
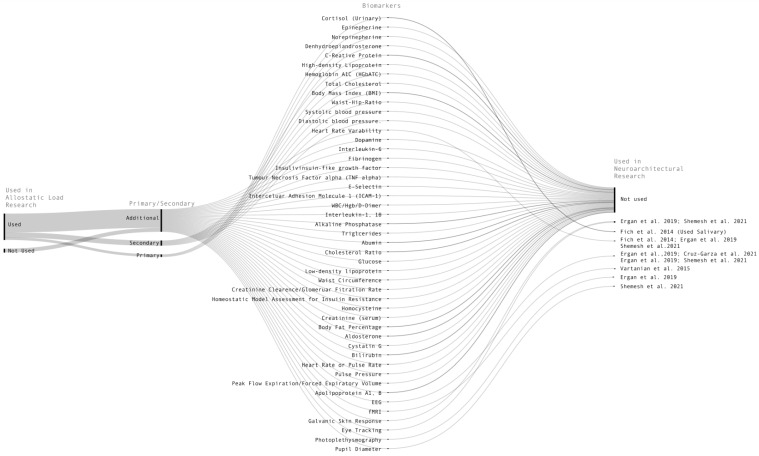
The alluvial diagram depicted above presents a visual representation of the interrelationship between biomarkers employed in the exploration of allostatic load [63] and those employed in the investigation of neuroarchitecture [11,12,13,14,15] Notably, none of the biomarkers used in the allostatic load studies were utilized in the neuroarchitecture studies. This observation implies that although the stress responses identified in neuroarchitectural research signify allostatic activity, they do not serve as confirmation of the existence of allostatic load.

Increasingly, there is a trend to develop algorithmic tools as a means of creating more refined and sophisticated statistical analysis techniques for computing allostatic load. The process of developing an allostatic load algorithm encompasses various considerations, such as the quantification of allostatic activity and the determination of the weightage for specific biological systems [77]. However, as noted by Carbone et al. (2022) [77], these algorithms are presently in their nascent stages of development, and as the theory of allostatic load progresses, it is expected that the implementation of more advanced and nuanced statistical analysis techniques will facilitate a more nuanced measurement of variables that contribute to allostatic loads, as well as enhance the effectiveness of interventions.

### 3.2. Clinimetric Assessment

While biomarkers provide objective measurements of neurophysiological function, Guidi et al., 2022 [78] (p. 12) observes that the “biological perspective does not allow for a comprehensive understanding of allostatic load and overload and related clinical phenomena, a substantial contribution has come from clinimetrics, the science of clinical measurements”. Clinimetrics expand on clinical biomarkers to incorporate patient-reported symptoms and physical signs of health outcomes as reported over time [79]. Therefore, clinimetrics combine both objective and subjective measures of human health [79]. The benefit of clinimetrics is that it allows the researcher to situate changes in physical symptoms within the context of the patient experience [79]. Clinimetrics has been applied in an attempt to contextualise and interpret the nature of the clinical variables reported regarding allostatic overload [79]. Subjective data are typically collected from surveys, questionnaires, or interviews that capture the patient’s interaction with an identified source of distress, and data on exhibited symptoms such as difficulties falling asleep, restless leg syndrome, or impairment in social or occupational functioning [80].

For example, Jung et al. (2014) [62] conducted a study to examine the potential influence of indoor environmental quality on allostatic load and its ability to predict sick building syndrome. The sample consisted of 115 office workers from 21 offices who completed self-reported questionnaires on sick building syndrome and provided 11 biomarkers indicative of their allostatic load. The questionnaire included personal information such as age, gender, working years, weekly working hours, and time spent in the building, as well as questions about symptoms experienced inside the building during the past four weeks. The findings indicated a correlation between CO_2_ levels and the neuroendocrine biomarkers, while illumination was found to be associated with the metabolic biomarkers. Furthermore, the study revealed that the neuroendocrine and metabolic systems were associated with the risks of sick building syndromes. The study concluded that indoor environmental quality significantly influences allostatic load and that, in turn, allostatic load can be a predictor for reporting sick building syndrome.

Clinimetrics can also encompass outcome-based evaluations which track health outcomes associated with allostatic overload over time [79]. If a specific health outcome, such as the onset of cardiovascular disease or depression, is used as an allostatic load indicator, repeated assessments of these outcomes can be conducted at multiple time points to track temporal changes. For instance, several studies have investigated the correlation between increased allostatic load and the incidence of multiple diseases associated with high morbidity and mortality rates, such as cardiovascular diseases, diabetes, musculoskeletal disorders, neurological disorders, and cancer [78]. By evaluating the prevalence and severity of these outcomes, researchers can infer the degree of allostatic overload experienced by an individual. As allostatic load theory advances, these outcome-based evaluations, along with other more refined statistical techniques, will likely enhance our understanding of the factors contributing to allostatic overload and inform more effective interventions.

Moreover, latent profile analysis (LPA) is a statistical approach that can potentially complement conventional methods of measuring allostatic load in clinimetric research [81]. LPA is a model-based approach that assumes that the observed variables are indicators of the unobserved latent variables [82]. The technique involves estimating a model that specifies the number of latent profiles or classes, as well as the relationships between the latent variables and the observed variables [82]. Application of LPA to allostatic load may involve examining the accumulation of stress and the resulting wear and tear on the body over an extended period of time. For example, a study might assess exposure to stressors experienced in childhood, adolescence, and adulthood and examine how these stressors contribute to allostatic load in later life. The application of this approach could be advantageous in exploring the effects of exposure to specific stress-inducing factors that may contribute to the likelihood of developing conditions associated with allostatic overload in the future [81].

### 3.3. Limitations

Measuring allostatic load is a complex process that involves assessing the impact of chronic stress on multiple physiological systems. While both biomarker-based and clinimetric assessments have been used to measure allostatic load, they have several limitations. One of the primary challenges associated with measuring allostatic load is the cross-sectional nature of most studies conducted to date [77]. Increasingly, however, studies have worked to incorporate longitudinal cohorts [83,84,85]. Longitudinal studies facilitate acquisition of multiple biomarker samples over extended periods of time, which allows for the temporal sequencing of allostatic load relative to other health-related variables [77].

The second limitation of measuring allostatic load is the lack of consensus on standardised measurement methods [77]. This variability is apparent in the use of different biomarkers, combinations of biomarkers, and clinimetric assessment techniques across studies [77,79]. Consequently, it is difficult to compare results across studies, and the measure’s validity and reliability are limited. For example, as discussed in Section 3.1, while some studies adopt uniformly assigned weightage to biomarkers, others prefer percentile-based approaches to measure allostatic load [72]. This lack of consistency highlights the need for the development of standardised methods of measuring allostatic load that can facilitate cross-study comparisons and enhance the validity and reliability of the measurements.

The third difficulty of measuring allostatic load is the limited scope of biomarkers [86]. Although allostatic load is conventionally measured using a combination of biomarkers that reflect various physiological systems, such as the cardiovascular, metabolic, and immune systems, these biomarkers may not capture the entirety of physiological responses to stress, including those associated with the neuroendocrine and inflammatory systems. Biomarkers instead offer a snapshot of physiological activity at the moment of sample collection [87]. In contrast, clinimetric measurements, while providing a comprehensive understanding of stress’s impact on health, may lack specificity and sensitivity and can be liable to methodological shortcomings [79]. Therefore, the use of a combined approach that integrates both biomarker-based and clinimetric assessments could potentially provide a more comprehensive and accurate assessment of allostatic load.

Fourth, the variability in responses to stress across populations may present a challenge in comparing allostatic load across individuals using biomarker-based or clinimetric assessments [88]. For example, while the decline in stress response efficacy with age poses a health hazard and contributes to the development and accumulation of age-related diseases in some individuals, others (particularly centenarians) appear to have developed a conservative stress response mechanism, which can mitigate stress overload at both the nuclear and mitochondrial DNA levels [89]. The variation in stress responses may be further complicated by preexisting conditions such as high blood pressure or diabetes [88].

Fifth, allostatic load is influenced by external factors, such as age, biological sex assigned at birth, and socioeconomic deprivation [90]. These factors can have a significant impact on measurement outcomes and therefore, may make it difficult to isolate the effect of specific factors on allostatic load [87]. However, a strength of the clinimetric approach is its capacity to contextualise the observed physiological responses, potentially capturing or providing insight into external factors [79]. A clinimetric methodology may be particularly useful when working to engage more integrative theories of stress, which consider stress to be mediated by cognitive appraisal, behavioural outcomes, and physiological mechanisms [91,92]. For example, an argument has been put forward suggesting that perseverative cognition in the forms of worry, rumination, and anticipatory stress should be taken into account due to their association with elevated levels of cardiovascular, endocrinological, immunological, and neurovisceral activity [93]. Additional research suggests that personality traits may also play a role in influencing an individual’s response to stress [94,95,96].

## 4. Application to Architectural Research

It is particularly interesting to note that neuroarchitecture studies that measure stress responses to architectural features do not employ the biomarkers associated with allostatic overload. As highlighted above, recent studies in the field of neuroarchitecture have considered the impact of certain architectural forms on human health [97]. Emerging from these studies is evidence that exposure to certain architectural forms, such as room proportions [14,15], wall curvature [12,15], and window arrangement and size [11,13,14], regularly provoke stress responses in humans without their conscious perception. These studies have used a combination of biomarkers to measure stress, particularly heart rate and heart rate variability [11,13,15], salivary cortisol [11], electroencephalogram (EEG) [13,14,15], galvanic skin response [13,15], photoplethysmography [13], eye tracking and pupil dilation [13,15], and functional magnetic resonance imaging (fMRI) [12].

On the other hand, from Section 3.1 we can extract the clinical biomarkers that are frequently used to measure allostatic load. The primary markers, identified above [63], are: cortisol, epinephrine, norepinephrine, and dehydroepiandrosterone. The secondary markers are: systolic and diastolic blood pressure, high-density lipoprotein, glycosylated haemoglobin, total cholesterol, body mass index, and waist-to-hip ratio. Additional markers include: C-reactive protein, interleukin-6, and fibrinogen. Although heart rate variability is commonly used to measure stress responses to architectural forms, heart rate variability is used to measure allostatic load in just 8.4% (n = 20) of the 238 studies reviewed by Beese et al. (2022) [63]. Similarly, while salivary cortisol has been used to measure the impact of architectural microstressors, urinary cortisol instead is a favoured biomarker of allostatic load research [63].

Additionally, other common measurement techniques used to measure stress responses to architectural forms such as EEG, photoplethysmography, galvanic skin response, eye tracking and pupil dilation, and fMRI are not identified as common measurements of allostatic load. Conversely, to the best of the authors’ knowledge, biomarkers frequently used to measure allostatic load (urinary cortisol, epinephrine, norepinephrine, dehydroepiandrosterone, systolic and diastolic blood pressure, high-density lipoprotein, glycosylated haemoglobin, total cholesterol, body mass index, waist-to-hip ratio, C-reactive protein, interleukin-6, fibrinogen, peak flow expiratory, and apolipoprotein A1, B) have not been used to measure stress-responses to architectural features.

While the measured stress responses to particular stress-inducing architectural forms [11,12,13,14,15] may suggest allostatic activity, it is not conclusive that those stress responses are resulting in allostatic overload without further empirical inquiry utilising the methods for studying allostatic load. It is possible that the individual could come to downregulate and return to homeostasis. It is equally possible that the repeated exposure to stress-inducing architectural forms could result in upregulation and allostatic overload. Therefore, we should not be too quick to assume a connection between stress-inducing architectural features and allostatic load. On the other hand, given what is known about both the impact of allostatic overloading on human health and stress-inducing architectural forms, the potential impact of chronic exposure to these stress-inducing architectural forms on human health should not be underestimated. If a connection between architectural form and allostatic load is in fact confirmed, architectural forms may contribute to higher mortality and morbidity levels.

Given the divergence in the biomarkers used to measure the impact of architectural forms on stress as compared to those used in the research on allostatic load, further empirical research is required. A future empirical study on the relationship between architectural form and allostatic load could begin by systematically identifying the architectural features which have been found to upregulate stress responses. To date, these features include: room proportions [14,15], wall curvature [12,15], and window arrangement and size [11,13,14]. The impact of these features on allostatic load could be assessed through a longitudinal public health study which evaluates changes in biomarkers of allostatic load: urinary cortisol, epinephrine, norepinephrine, dehydroepiandrosterone, systolic and diastolic blood pressure, high-density lipoprotein, glycosylated haemoglobin, total cholesterol, body mass index, waist-to-hip ratio, C-reactive protein, interleukin-6, fibrinogen, heart rate/pulse rate, pulse pressure, peak flow expiratory, and apolipoprotein A1, B [63].

These biomarkers may be used to construct a composite biomarker index or could be integrated with a clinimetric approach. This data would need to be evaluated in conjunction with data concerning confounding variables which are known to contribute to the development of increased allostatic load, including prolonged low socioeconomic status [73,98]; work-related conditions [18,21,23,99]; loss of individual resources [17,19,20]; complex childhood trauma [22,100]; lead exposure [55,56]; perception of pollution [57]; pollution [58]; dangerous traffic [59]; household crowding [60]; environmental riskscapes [61]; and poor indoor environmental quality [62] (see Section 3.3). A combined biomarker/ clinimetric approach may be appropriate in the field of neuroarchitecture where it is particularly difficult to isolate spatial features from external factors.

## 5. Conclusions

This paper examined, conceptually, the relationship between stress-inducing architectural features and allostatic overload by drawing on literature from neuroimmunology and neuroarchitecture. The studies reviewed from the field of neuroimmunology strongly suggest that chronic or repeated exposure to stress may produce a cascade of allostatic events which result in allostatic overload. While there is evidence from the field of neuroarchitecture that short-term exposure to particular architectural features produces acute stress responses, there is yet to be a study on the relationship between stress-inducing architectural features and allostatic load.

The paper went on to consider how such a study may be designed. Of particular interest was the observation that the clinical biomarkers used to measure stress in neuroarchitectural studies differ substantially from those used to measure allostatic load. Therefore, while the observed stress responses to particular architectural forms may indicate allostatic activity, further research is needed to determine whether these stress responses are leading to allostatic overload. Consequently, a discrete longitudinal public health study is advised, one which engages the clinical biomarkers of allostatic load. This study could also incorporate subjective data using a clinimetric approach.

The architectural features of interests were identified as: room proportions [14,15], wall curvature [12,15], and window arrangement and size [11,13,14]. The primary and secondary clinical biomarkers for measuring allostatic load are: urinary cortisol, epinephrine, norepinephrine, dehydroepiandrosterone, systolic and diastolic blood pressure, high-density lipoprotein, glycosylated haemoglobin, total cholesterol, body mass index, waist-to-hip ratio, C-reactive protein, interleukin-6, fibrinogen, heart rate/pulse rate, pulse pressure, peak flow expiratory, and apolipoprotein A1, B [63]. Finally, any such study would face certain limitations, particularly the lack of consensus on measurement techniques, the limited scope of biomarkers, and the potential influence of confounding variables that hinder the isolation of spatial variables. Despite these limitations, given that chronic inflammatory diseases are the leading cause of death worldwide, accounting for three out of five deaths [25], further research exploring the relationship between stress-inducing architectural forms and allostatic load is highly recommended.

## Figures and Tables

**Figure 1 ijerph-20-05637-f001:**
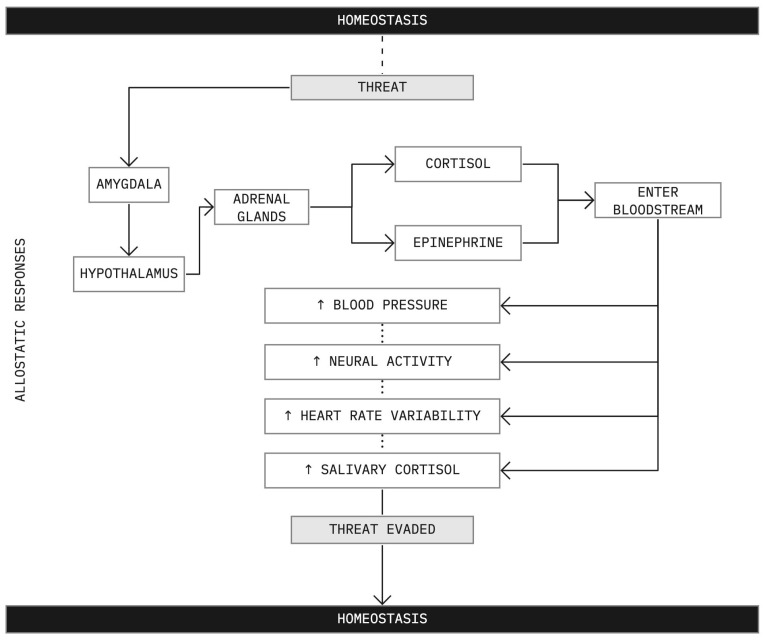
Overview of the allostatic response to a perceived stressor outlined above. The individual in question begins in a state of homeostasis. Once a perceived threat is registered, allostatic responses are engaged, which allows the individual to evade the threat and return to homeostasis.

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
