# Peer review of "Architectural Allostatic Overloading: Exploring a Connection between Architectural Form and Allostatic Overloading"

_ijerph, 2023, doi:10.3390/ijerph20095637_

Round 1

Reviewer 1 Report

This is a very interesting topic, and it may be an obvious phenomenon that architectural allostatic overloading has an effect on neural activity, especially for emotion, however, this article only gave a concept and research method or frame, not original research, it`s a beautiful work if you can test the hypothesis with data.

Author Response

Dear Reviewer, 

Thank you for your thoughtful review of this manuscript. The current version of the document has been substantially revised and improved as a result of your insightful comments and valuable feedback. 

The paper has been restructured and an in depth analysis of the methods used to measure allostatic load has been conducted and incorporated into the revised article. These revisions provide the reader with practical insight into how to measure the proposed relationship between stress-inducing architectural forms and allostatic load. A particularly interesting point that emerged from this revision process was that the neuroarchitectural studies that measure stress responses to architectural forms used markedly different biomarkers to those that measure allostatic load. Consequently, the revised manuscript advises that, given the different measurement techniques, while the research on stress-inducing architectural forms may be capturing allostatic activity, the methods used are not appropriate for measuring the long-term impacts of these architectural forms on allostatic load. Therefore, the paper now concludes that further research is needed to determine whether these stress responses are leading to allostatic overload, using the methods highlighted in the revised manuscript.  

While the paper does not incorporate original data from empirical research, the insights provided by the interdisciplinary review are - it is hoped - sufficiently novel and capable of informing future research in an emerging field. It is hoped that the revisions now provide a more robust conceptual framework for understanding the connection between stress-inducing architectural features and allostatic load, highlight certain issues with assuming a connection between stress-inducing architectural features and allostatic load given the differing approaches to measurement, and provides insight into how a future longitudinal public health study into the relationship between architecture and allostatic load could be conducted.

Thank you once again for your comments, the manuscript has substantially benefited from your considered insights. Please note that I have carefully considered each of your comments and have attempted to address them all. A more detailed point-by-point response is attached below.

Kind regards.

Reviewer 2 Report

In the review, the author discuss recent literature that associates specific aspects of architecture with several acute stress responses in humans.  The author then suggest that long-term exposure to such environmental features may lead to allostatic overloading and proposes a new term “architectural overloading” for this hypothetical construct.  The review is nicely presented and covers significant recent findings.  In this reviewer’s opinion, however, the hypothetical aspects of the manuscript lack any experimental backing and can be truncated.

The author should address the following comments:

1.     The author clearly present results from several studies describing stress responses to different environments.  However, they do not discuss the actual experimental design in those studies. What controls were used? How do those authors separate response to novelty rather than specific environmental features as the drivers of the stress responses?  These are significant issues that should be discussed in this review.

2.     While it is reasonable to suggest that long term exposure to detrimental environments could lead to allosteric overloading, it does not seems appropriate to propose  “architectural overloading” as a new terminology when, as the authors note, there is no evidence to support this hypothesis. It would be valuable to suggest this as a possibility and then propose a discrete set of studies that would provide evidence for such a long-term phenomenon in humans.

Author Response

(The authors gave the same response as above.)

Reviewer 3 Report

This short communication looks mainly like a kind of review, with a focus primarily on the examination of (immediate) neurophysiological responses to the built environment (Introduction 1st paragraph, line 6). This is despite the author's claim to have a main thrust intended to be directed towards studying chronic exposure to stress-inducing architectural forms (Abstract line 13).

This raises two main questions:

(I) Originality in the submitted manuscript is sought mainly in the author's claim to study 'the impact of long-term or chronic exposure to stress-inducing architectural features on human health... yet to (be) examine(d)' (Abstract). The part of the manuscript addressing what should be the principal part of the work is however a relatively superficial account on pages 6 and 7. I cannot find any account of for instance empirical results of the author's about the specific topic of chronic exposure and its effects. Instead, the author gives again a review; at best making some "suggestions" (last paragraph of section 3).

That long-term exposure to stress can lead to chronic effects will be no surprise to any reader and is readily mentioned in the literature on stress, e.g: Pretty J, Peacock J, Sellens M, Griffin M. 2005, The mental and physical health outcomes of green exercise, International Journal of Environmental Health Research 15, 319–37. [PubMed: 16416750].

One key Allostasis paper early on by McEwen 1998 describes Stress research as: "understanding of psychological, physiological and behavioural mechanisms leading from stress exposure to stress response and the development of stress-related health problems".

McEwen BS: Stress, Adaptation, and Disease: Allostasis and Allostatic Load. Ann N Y Acad Sci 1998, 840(1):33-44.

Guidi et al. 2021 [28] has recently given a systematic review of Allostatic Load and Its Impact on Health, where Allostatic Load is explicitly 'the cumulative burden of chronic stress and life events'.

I look forward to a manuscript from the author where perhaps the Introduction could be formed starting with section 3 of the present manuscript, but then containing the results of original investigations of chronic stress.

(II) We are promised (middle of Abstract) "a conceptual framework for understanding the relationship between chronic exposure to stress-inducing architectural forms and chronic inflammation". The author appears, in making his review (sections 1 & 2) of neurophysiological responses to the built environment, to present a rather restricted version of the promised "conceptual framework" with a particular bias towards "Stress responses ... empirically measured" (Section 1, 2nd paragraph) and explained (section 2 , figure 2 using clinical biomarkers.

But, if I understand correctly, stress response is not merely registered with clinical biomarkers. Indeed, stress responses can be both affective, behavioural or biological. [Cohen S, Kessler RC, Gordon LU: Strategies for measuring stress in studies of psychiatric and physical disorders. In: Measuring stress: a guide for health and social scientists. edn. Oxford; New York, N.Y: Oxford University Press; 1995: 3-26].

Some researchers may consider affective and behavioural responses to be "less objective" and "more difficult to measure" than biomarkers. On the other hand, arguably, a person's stress experience will be dominated by these affective and behavioural effects, rather than changes in their biomarkers. There is, however, a substantial body of research employing the psychometric measurement theory of Rasch which can provide objective measures of affective and behavioural constructs: both of the level of stress presented by a stressor (e.g. an architectual feature) and the sensitivity of a person to become stressed. These stressor and person-related attributes should be able to be modelled separately in terms of explanatory variables, such as in construct specification equations.

An example of recent stress research employing Rasch measurement theory is the work of Hadžibajramović E, Ahlborg G, Håkansson C, Lundgren-Nilsson Å, Grimby-Ekman A: Affective stress responses during leisure time: Validity evaluation of a modified version of the Stress-Energy Questionnaire. Scand J Public Health 2015, doi: 1403494815601552, first published on Sept. 21, 2015.

The author therefore needs to complement the "biomarker" focus in his conceptual framework with accounts covering affective and behavioural responses to stress with modern measurement theory.

Overall, to make the manuscript acceptable, I need to see convincing responses to these two main questions (I) and (II)

Author Response

(The authors gave the same response as above.)

Round 2

Reviewer 1 Report

Urbanization is highly advanced, and people exposed to chronic or repeated stress may produce a cascade of allostatic events which result in allostatic overload. This paper examined, conceptually, the relationship between stress-inducing architectural features and allostatic overload by drawing on literature from neuroimmunology and neuroarchitecture, which is an interesting topic between neuro and architecture, and hope your original idea can be realized sooner.

Reviewer 2 Report

Thanks for the author for taking all the reviewers' comments seriously and re-submitting a significantly improved and valuable paper.  This reviewer has no further comments.

Reviewer 3 Report

No further comments - thank you, author, for responding to my earlier comments and suggestions.